Enhancing knee osteoarthritis detection with AI, image denoising, and optimized classification methods and the importance of physical therapy methods

Bugday Burak 1
http://orcid.org/0000-0001-5071-4616 Bingol Harun 2 harun.bingol@ozal.edu.tr
http://orcid.org/0000-0003-1866-4721 Yildirim Muhammed 3
http://orcid.org/0000-0002-3513-0329 Alatas Bilal 4
1 Therapy and Rehabilitation, Inonu University , Malatya , Turkey
2 Software Engineering, Malatya Turgut Ozal University , Malatya , Turkey
3 Computer Engineering, Malatya Turgut Ozal University , Malatya , Turkey
4 Software Engineering, Firat (Euphrates) University , Elazig , Turkey
Asif Muhammad
Electronic publication date: 2025 Mar 25
Publication date: 2025
Volume: 11
Electronic Location ID: e2766
Received 2024 Dec 5; Accepted 2025 Feb 25
Copyright: © 2025 Bugday et al.
Copyright year: 2025
Copyright holder: Bugday et al.
License: This is an open access article distributed under the terms of the Creative Commons Attribution License, which permits unrestricted use, distribution, reproduction and adaptation in any medium and for any purpose provided that it is properly attributed. For attribution, the original author(s), title, publication source (PeerJ Computer Science) and either DOI or URL of the article must be cited.
License URL: https://creativecommons.org/licenses/by/4.0/

Keywords: Artificial intelligence, CNN, Gauss, Knee osteoarthritis, Machine learning

Funding: The authors received no funding for this work.

==============================
Osteoarthritis (OA) is considered one of the most challenging arthritic disorders due to its high disease burden and lack of effective treatment options that can change the course of the disease. Knee osteoarthritis (KOA) reduces people’s quality of life and shortens their daily activities. Therefore, early detection of KOA dramatically impacts patients’ quality of life. This study developed an artificial intelligence-supported system to detect KOA. In the developed system, firstly, the images in the original dataset were denoised with a Gaussian filter. Then, feature maps were extracted from both the original and Gaussian applied datasets with the DenseNet201 selected from eight different pre-trained models, and these two feature maps were concatenated. In this way, it is aimed to bring together different features of the same image. Then, feature selection was made using the neighborhood component analysis (NCA) method for the developed system to produce more successful results, and the optimized feature map was classified into six different classifiers. As a result, a high accuracy rate of 85% was achieved in the proposed model. This value is promising for the automatic diagnosis of KOA with computer-aided systems. As a result, a high accuracy rate of 85% was achieved in the developed system of the support vector machine (SVM) classifier. The proposed model was more successful than the other models used in the study.

Introduction

Numerous specific risk factors, such as obesity, advanced age, and female gender, as well as concomitant illnesses, such as stroke, hypertension, peptic ulcers, anxiety, depression, diabetes, and obesity, are linked to osteoarthritis (OA), a chronic, complex disorder. This raises the expense of treatment as well as the difficulty of managing the disease. Given the increasing prevalence of adult obesity and the aging of the population in recent decades, concerns about the future burden of the disease arise (Savvari et al., 2024). Knee osteoarthritis (KOA) is a chronic degenerative disease particularly prevalent in individuals over 60 and in women. Major risk factors for OA include genetic mutations, obesity, trauma, biomechanical factors, and hormonal imbalances. Complications of OA, such as cartilage injury and synovitis, are the main causes of disability in adults. The disease affects more than 250 million people worldwide and poses a significant economic burden on countries (Xu, Zhang & Fang, 2024).

The diagnosis of KOA is usually based on symptoms such as knee pain, morning stiffness, and limited range of motion, as well as physical signs such as pain with joint movement, crepitus, joint tenderness, and bone enlargement. This approach is considered the most reliable method to confirm KOA. Using diagnostic criteria recommended by organizations such as the American College of Rheumatology, the Chinese Orthopaedic Association, and the European League Against Rheumatism is recommended. According to the 2022 National Institute for Health and Care Excellence guidelines in the United Kingdom, KOA can be diagnosed with confidence if individuals over 45 have joint pain during activity and morning stiffness lasting less than 30 min. However, radiological imaging and laboratory tests are usually only used when atypical symptoms (e.g., prolonged joint stiffness or pain at rest), suspicion of a different disease (e.g., calcium pyrophosphate crystal deposition), or a sudden worsening of symptoms (Zhu, Qu & He, 2024). A thorough medical history and physical assessment of the patient are important to understand the effects of KOA on the person’s functionality, quality of life, social relationships, work, leisure activities, and sleep (Duong et al., 2023). In the early stages, KOA patients often experience symptoms such as knee joint swelling, pain, effusion, and decreased proprioception. These symptoms not only reduce joint mobility and stability but also have a significant impact on daily activities, work productivity, and mental well-being. Consequently, early nonsurgical interventions are crucial for managing KOA in patients (Lai et al., 2021).

OA is considered one of the most challenging arthritic disorders due to its high disease burden and lack of effective treatment options that can modify the course of the disease. Symptomatic early-stage OA of the knee is the focus of this review, and the importance of early diagnosis cannot be overstated (Emami et al., 2023). If early-stage cases are identified and diagnosed in primary care, health care providers can proactively and dramatically reduce the disease burden by effective management, including structured education, exercise, weight management, and addressing lifestyle risk factors for disease progression. Ongoing efforts are being made to identify patient groups with symptomatic early-stage knee OA using established classification standards. Identifying molecular and imaging biomarkers that can predict disease risk and progression, along with these classification criteria, will pave the way for well-designed clinical trials, facilitating interventional trials and discovering and validating cellular and molecular targets for new therapies. When it comes to the economical treatment of early-stage, symptomatic knee OA, consideration of treatment strategies, targeted outcomes, and ethical issues is also crucial. Future work requires an interdisciplinary and sustainable international collaboration involving all relevant stakeholders (Mahmoudian et al., 2021).

Early KOA diagnosis may offer a “window of opportunity” to restore joint homeostasis and stop the disease process early, enabling earlier disease management. Despite the widespread belief that KOA is difficult to manage, some tools are available to manage and reduce its burden, especially in the early stages. These tools include but are not limited to, education and exercise programs, injury prevention, development of coping strategies, and adding pain medications when necessary. Early intervention also allows personalized lifestyle changes that encourage exercise and weight control (Mahmoudian et al., 2018). Age, obesity, and prior knee injuries are significant risk factors for OA. Treatment strategies are diverse and prioritize nonpharmacological measures such as patient education, rehabilitation exercises, and weight management, with pharmacological interventions considered adjunctive. Intra-articular injections and surgical options are considered when conventional therapy is inadequate. OA is a globally dominant cause of disability, characterized by a complex etiology and profound impacts on individuals’ quality of life. Early, proactive treatment focusing on nonpharmacological interventions is the cornerstone of treatment aimed at relieving symptoms and improving joint function. This comprehensive review highlights the need for early diagnosis, personalized treatment plans, and integration of rehabilitation assessments to optimize patient outcomes. Further research is needed to develop prevention strategies and improve treatment outcomes for patients with KOA (Zhu, Qu & He, 2024).

Current radiographic assessment scales in OA are based on examining changes in X-ray images, usually based on the Kellgren-Lawrence (KL) grading. However, this approach delays the diagnosis of OA because bone changes only appear in advanced stages. In addition to X-ray, other imaging techniques such as magnetic resonance imaging (MRI) allow earlier detection of OA onset using soft tissue biomarkers. Deep learning (DL) approaches and classical methods are the two main categories of classification and segmentation models utilized in OA assessment. Visual evaluation of radiographic images is typically used in clinical practice to determine the severity of OA, which takes time in big data sets and introduces interobserver variability (Yeoh et al., 2021). Computer-aided KOA diagnosis, which attempts to lessen diagnostic ambiguities brought on by human mistake, is one example of how recent physiotherapy research have embraced artificial intelligence and come to recognize the importance of deep learning in the medical sector (Hayashi, Roemer & Guermazi, 2019; Davids, Lidströmer & Ashrafian, 2022). Imaging methods play a vital role in the early diagnosis of OA. Deep learning has been successfully used in image-based knee OA diagnosis. In particular, Convolutional neural networks (CNN) are the most widely used deep learning architecture in medical image processing. These networks have demonstrated high accuracy rates in classification, detection, and segmentation tasks in knee OA diagnosis. While traditional methods can be time-consuming, deep learning techniques have provided speed and accuracy on large datasets (Wirth et al., 2021).

Assessing and treating physical injuries and disabilities is the focus of the vital healthcare profession of physiotherapy. The potential of artificial intelligence (AI) in physiotherapy to enhance the precision and efficacy of clinical judgment and treatment results has garnered much interest. However, it is still a highly creative sector that uses these methods (Nogales, Rodríguez-Aragón & García-Tejedor, 2024). Technological advancements have been aggressively integrated into the physiotherapy industry. Analyzing signals from various treatments and interventions using various assessment techniques is now usual practice. This procedure makes it easier to track patients’ disease progression and regression and monitor them continuously. Large volumes of data are thus produced, necessitating the use of suitable analytic techniques. This circumstance emphasizes the necessity of incorporating cutting-edge technical advancements like artificial intelligence (AI) into physical therapy. The use of AI presents the potential for creating more individualized and successful treatment plans. The branch of computer science known as artificial intelligence examines and deciphers the processes that lead to the development of appropriate human behavior. The ultimate objective is to use different techniques to replicate these behaviors in machines (Russell & Norvig, 2016).

In this study, a novel method for automatically diagnosing and grading KOA using plain radiographs is presented. Unlike previous articles, our system uses specific disease-related features (e.g., bone shape, joint space, etc.,) similar to those used in clinical practice. Moreover, it demonstrates that it can perform at a near-human level when compared to previously published approaches. In conclusion, we think there are a number of advantages to the suggested strategy. First, it will facilitate a quicker diagnosis for individuals experiencing knee discomfort, which will assist OA patients undergo proactive rehabilitation. Second, it will benefit the healthcare service in general by reducing the costs of routine work. Although the present study focuses on OA, the model we developed can systematically assess OA and other problems related to the knee condition. This can cover different situations, such as monitoring the recovery process after ligament surgery or observing changes in the joint. In addition, the implementation of early rehabilitation protocols in OA patients suggests that chondroprotective (cartilage-preserving) treatment options could be used by directly manipulating the microenvironment of the damaged joint. This could make the treatment process more proactive and effective.

Innovations of the study and contributions to literature

KOA is a disease that occurs due to the wear of the cartilage in the knee joint. Cartilage prevents the bones in the joint from rubbing against each other, allowing them to move comfortably. However, in the case of calcification, this cartilage thins and deteriorates over time, and the bones in the joint begin to rub against each other. This leads to pain, swelling, stiffness, and limited movement.

Therefore, detecting knee calcification with computer-aided systems at an early stage is vital. Early detection of KOA with computer-aided systems will allow early initiation of treatment.

In this study, eight pre-trained models were used to detect KOA. These models were used for feature extraction, and the obtained features were classified in six classifiers. As a result, the highest performance was obtained in the support vector machine (SVM) classifier in the features extracted with the DenseNet201 architecture.

A hybrid model was also developed for the early detection of knee calcification. The DenseNet201 architecture was used as the base in the developed model. In addition, a Gaussian filter was applied to the original data set to eliminate noise in the images.

The DenseNet201 model extracts features from the original and Gaussian-applied data sets. Using the obtained feature maps, different features of the same image are blended. The optimized feature map is then classified in the SVM classifier after NCA is performed on the merged feature map to speed up and improve the model’s performance.

The proposed model achieved competitive results and an accuracy value of 85%.

Organization of paper

Related Studies on Measuring KOA Severity Autonomously, Theoretical Background, Application Results, Discussion, and Conclusion sections are reviewed in the rest of the article.

Related studies to measure knee osteoarthritis (KOA) severity autonomously

KOA is a serious health problem that significantly negatively affects individuals’ quality of life. Therefore, automatic detection of KOA with computer-aided systems is of great importance in terms of increasing the effectiveness of diagnosis and rehabilitation processes. Various studies are in the literature in this field, and intensive research is being carried out on the development and improvement of such systems.

Deep learning has been utilized extensively lately for various applications, including natural language processing (NLP) and computer vision (CV). CNNs have demonstrated superior representation and hierarchical learning capabilities in several applications. They are also highly effective in the analysis of medical imaging. An image classification task that is very appropriate for CNNs is the automatic identification of the severity of KOA (Litjens et al., 2017).

Zhang et al. (2020) focused on estimating knee joint localization and KL degree with Residual neural network (ResNet) and Convolutional Block Attention Module (CBAM) to detect KOA severity automatically. In this study, ResNet-18 and ResNet-34 architectures were used, and it was stated that CBAM, in particular, contributed to achieving high accuracy by detecting the regions of interest in radiographic images more accurately. The accuracy rate of the model was reported as 74.81% (Zhang et al., 2020). By removing bone masks from the bone-cartilage complex, Lee, Hong & Kim (2018) carried out a cutting-edge bone-cartilage segmentation procedure to remove cartilage. Segmentation masks in several planes were averaged, and majority voting was used to accomplish 2.5D segmentation. For the femoral and tibial cartilages, the suggested BCD-Net obtained DSC values of 98.1% and 83.8%, respectively (Lee, Hong & Kim, 2018). Pedoia et al. (2019) segmented the patellar cartilage in 1,478 participants’ TSE images using U-Net CNN. To ascertain whether cartilage lesions were present according on the musculoskeletal radiologist’s judgment, the segments were examined using a specialized classification CNN. With 80% sensitivity and specificity, the deep learning approach obtained an AUC of 0.88 on the test dataset of 222 patellar cartilages (Pedoia et al., 2019). In people with or at risk for KOA, Tiulpin et al. (2018) employed a deep Siamese CNN architecture to identify the degree of OA in posterior-anterior knee radiographs using the KL system. Experienced radiologists used the OAI database to train the CNN on 6,852 knees. The deep learning approach obtained a multiclass accuracy of 67% and a weighted kappa coefficient of 0.83 on the test dataset of 3,000 knees. Additionally, it identified radiographic knee OA with KL grade 2 or higher with an AUC of 0.93 (Tiulpin et al., 2018). In this study, CNNs are also highlighted. Antony et al. (2016) presented a deep learning-based method to identify the degree of KOA in posterior-anterior knee X-rays using the KL system in individuals who already had or were at risk for developing the condition. A total of 6,224 knees were used to train the VGG-M-128 CNN, and the OAI database’s comments from seasoned musculoskeletal radiologists were consulted. The deep learning approach outperformed a Wndchrm non-CNN-based classifier with a multiclass accuracy of 67% in assigning the same KL grade as the reference standard in the test dataset of 2,668 knees. It may be possible to increase the performance of the study with more data (Antony et al., 2016). In their study for the detection of KOA, Yildirim & Mutlu (2024) performed feature extraction using textural and CNN-based methods. They eliminated unnecessary features from the feature maps they obtained with the NCA method and classified the optimized feature map in different classifiers. The researchers achieved an accuracy value of 83.6% in the proposed model (Yildirim & Mutlu, 2024).

Teoretical background

This section of the article explains the feature extraction methods used, the Gaussian method used to remove noise from the image, the NCA method used for dimension reduction, the data set, and the proposed model.

Technological infrastructure

The publicly available digital knee X-ray dataset was used during the experiments (Kaggle, 2020). This dataset has five classes. The dataset has 514, 477, 232, 221, and 206 images in normal, doubtful, mild, moderate, and severe, respectively. The Kellgen and Lawrence methods were used to grade the knee images in the dataset, and experts in the field classified them. This article used state-of-the-art deep architectures to detect KOA in wireless capsule endoscopy images.

The AlexNet architecture was proposed by Alex Krizhevsky, who gave its name to this architecture. This 25-layer architecture won the Imagenet competition 2012 using a softmax activation function (Krizhevsky, Sutskever & Hinton, 2012). The DenseNet201 architecture was developed by Huang et al. (2017). It is structurally very similar to the ResNet architecture. The difference between this architecture and the ResNet architecture is that the activation functions are combined instead of added to the following layers (Huang et al., 2017). Tan & Le (2019) developed the EfficientNetb0 architecture. The main feature of this architecture is that the depth, width, and resolution dimensions are scaled equally using a fixed coefficient. The GoogleNet architecture was developed by Szegedy et al. (2015). This architecture was designed by using parallel-connected modules instead of adding many layers on top of each other to reduce the memory usage of the network and the possibility of over-learning the network (Szegedy et al., 2015). The InceptionV3 architecture was developed by Szegedy et al. (2015) just like the GoogleNet architecture. The main feature of this model is using inception modules to increase computational efficiency without decreasing the classification accuracy (Szegedy et al., 2016). MobileNetV2 architecture was developed by Howard et al. (2017). This CNN architecture is optimized primarily for portable devices with low computational capacity (Howard et al., 2017). ResNet50 architecture was developed by He et al. (2016). This architecture uses residual connections to eliminate vanishing gradients and degrading problems. In this way, it presents a more efficient network structure (He et al., 2016). ShuffleNet architecture is also a CNN architecture developed for devices with low computational capacity, like MobileNetV2 architecture. This architecture can also be used effectively in real-time image processing applications (Zhang et al., 2018).

Generally, CNN architectures can achieve high-accuracy classification on 2D images. However, the architectures used in this study were taken as a basis, aiming to increase the accuracy values by trying to improve them. For this purpose, the Gaussian method was used to reduce the noise in the dataset’s WCE images. The Gaussian method is a method that generally tries to eliminate the noise in the image by replacing each pixel value in the image with the weighted average of the neighboring pixels (Ito & Xiong, 2000). The study used different classification algorithms accepted in the literature to classify the feature maps obtained. These algorithms are SVM (Chapelle, Haffner & Vapnik, 1999), k-nearest neighbors (KNN) (Weinberger, Blitzer & Saul, 2005), neural network (NN) (Wang & Wang, 2003), naive Bayes (NB) (Rish, 2001), logistic regression (LR) (Kleinbaum et al., 2002), decision tree (DT) (Quinlan, 1996).

In the study, feature selection was performed with the neighborhood component analysis (NCA) so that the models can work faster and produce more effective results. NCA uses a truth function during learning. This truth function rearranges the distances between classes and aims to make the examples from the same class closer and the examples from different classes farther. Thus, a feature space is obtained in which the data points are better grouped and more distinct classes are created (Goldberger et al., 2004).

Proposed model

In the proposed study for the detection of knee arthritis, WCE images were used. DenseNet201 was used as the base in the proposed model because it achieved the highest performance among the eight architectures used in the article. While the size of the train feature map obtained in the original dataset with the DenseNet201 architecture was 1,321 × 1,000, the size of the test matrix was 329 × 1,000. A Gaussian filter was applied to the original images in the proposed model to obtain high-performance metrics. In this way, it aimed to eliminate the image noise and obtain more successful features. Then, the feature maps of the improved images obtained by applying Gaussian were extracted with the DenseNet201 architecture. As in the original dataset, the size of the train feature map was 1,321 × 1,000, while the size of the test matrix was 329 × 1,000. Then, the feature maps obtained from the original and Gaussian-applied datasets were concatenated. At this stage, the aim was to bring together different features of the same image. At this stage, the size of the train feature map was 1,321 × 2,000, while the size of the test matrix was 329 × 2,000. The NCA method was used to make the developed system work faster and obtain more successful performance measurement metrics. At this stage, the size of the train feature map was 1,321 × 600, and the size of the test matrix was 329 × 600. The block diagram of the proposed system is presented in Fig. 1.

Figure 1 Flow diagram of the proposed model.

The proposed system’s performance was also compared with the results obtained by classifying the features using eight different pre-trained models in six different classifiers.

Application results

The results for the early detection of KOA with computer-aided systems were obtained in the MATLAB environment on a computer with an i7 processor, 32 GB memory, a 6 GB graphics card, and Windows 11 operating system. The models’ performance was evaluated using measures such as F1-score, Matthews correlation coefficient, accuracy, sensitivity, and specificity (Saito & Rehmsmeier, 2015).

Results obtained on pre-trained Models

In order to compare the performance of the proposed model for the detection of KOA with computer-aided systems, the results of eight different pre-trained models are presented. The most successful architecture among these models was DenseNet201, while the least successful architecture was EfficientNetb0. Confusion matrices represent classes 1-Normal, 2-Doubtful, 3-Mild, 4-Moderate, 5-Severe. Pre-trained models were run for 15 epochs and 1,230 iterations. All models were obtained using the same training parameters to compare the results successfully. Also, when the features obtained from the models were given to the classifiers, the parameters of the classifiers were kept constant. Table 1 shows the confusion matrices that were acquired in pre-trained models.

Table 1 Confusion matrices of CNN architectures.

	AlexNet	GoogleNet	
True class	1	85	12	5	1		True class	1	88	8	7			
2	22	54	17	2		2	18	52	24		1	
3	2	25	9	2	8	3	2	24	11	5	4	
4		1	2	33	8	4		2	4	28	10	
5	1	3		1	36	5		5		1	35	
	1	2	3	4	5		1	2	3	4	5	
	Predicted Class		Predicted class	
ShuffleNet	EfficientNetb0	
True class	1	69	22	11	1		True class	1	29	32	28	14		
2	28	49	16	1	1	2	26	32	27	8	2	
3	3	23	11	6	3	3	10	23	2	7	4	
4	1	1	2	32	8	4	1	4		30	9	
5		6	1	2	32	5	1	4		1	35	
	1	2	3	4	5		1	2	3	4	5	
		Predicted class		Predicted class	
ResNet50	InceptionV3	
True Class	1	79	16	7	1		True Class	1	73	17	13			
2	21	52	18	2	2	2	15	55	23	1	1	
3	2	25	8	5	6	3	2	28	8	4	4	
4		3	5	28	8	4		1	1	31	11	
5		3	1		37	5	1	2		2	36	
	1	2	3	4	5		1	2	3	4	5	
		Predicted class		Predicted class	
MobileNetV2	DenseNet201	
True class	1	84	13	6			True class	1	83	13	7			
2	19	43	28	4	1	2	21	52	22			
3	2	21	14	3	6	3	4	16	15	3	8	
4	1	2	1	30	10	4		1	1	31	11	
5		3		2	36	5		2	1	1	37	
	1	2	3	4	5		1	2	3	4	5	
	Predicted class		Predicted class	

Examining Table 1, it is evident that DenseNet201 is the most effective model, with an accuracy rate of 66.26%. The DenseNet201 architecture correctly classified 218 of 329 test images while incorrectly classifying 111. The EfficientNetb0 architecture was the least successful in detecting knee arthritis. The EfficientNetb0 architecture correctly classified 128 of 329 test images while incorrectly classifying 201. The accuracy value of this model was 38.91%.

Classification of feature maps obtained in DenseNet201 architecture using classifiers

Table 2 shows the accuracy values derived from classifying the feature maps from eight separate models in six different classifiers to compare the performance of the suggested model.

Table 2 Classification of features obtained with CNN architectures in classifiers (%).

	DT	LR	KNN	NB	SVM	NN	
AlexNet	50.7	65.3	63.2	50.0	72.4	69.2	
DenseNet201	53.6	70.8	72.8	58.1	78.4	73.5	
EfficientNetb0	53.2	68.4	72.7	55.8	77.1	75.3	
GoogleNet	51.5	64.8	68.8	52.3	72.9	68.9	
InceptionV3	53.6	68.1	71.6	57.6	76.2	69.9	
MobileNetV2	50.8	69.3	74.2	58.7	79.3	73.0	
ResNet50	55.2	67.2	73.5	56.4	77.1	72.4	
ShuffleNet	53.6	68.0	72.8	54.2	76.7	72.3	

The DenseNet201 architecture was the foundation for the suggested model since it produced the best performance rate among the pre-trained models. The features acquired in eight distinct models were optimized using the NCA approach, and they were subsequently categorized into six different classifiers to assess the effectiveness of the suggested model. One thousand features were acquired for every image. Then, 600 features were selected for each image from these features with the NCA method. In the experiments conducted at this stage, the cross-validation value was selected as five. The obtained accuracy values are presented in Table 3.

Table 3 Classification of features obtained with CNN architectures in classifiers after applying NCA (%).

	DT	LR	KNN	NB	SVM	NN	
AlexNet	52.2	66.3	71.3	51.7	74.6	70.4	
DenseNet201	55.1	71.3	73.2	59.0	79.6	74.8	
EfficientNetb0	55.1	70.7	74.7	56.8	77.9	74.7	
GoogleNet	52.3	67.0	71.9	52.2	74.3	69.6	
InceptionV3	51.5	70.2	73	59.6	75.9	68.7	
MobileNetV2	50.3	69.8	74.5	59.3	78.3	72.2	
ResNet50	54.5	70.4	73.7	56.5	78.2	73.5	
ShuffleNet	55.5	69.5	75.3	54.9	77.3	72.6	

When Table 2 is examined, the Cubic SVM classifier achieved the highest accuracy value, 79.6%, after the features obtained with the DenseNet201 architecture were optimized with NCA. Table 4 gives the confusion matrices obtained by classifying the features obtained in the DenseNet201 architecture into six different classifiers.

Table 4 Confusion matrices of DenseNet201-NCA-classifiers.

SVM	DT	
True class	1	440	66	7	1		True class	1	333	134	32	11	4	
2	59	375	34	5	4	2	135	254	53	18	17	
3	7	55	144	16	10	3	20	61	86	35	30	
4		7	19	188	7	4	14	17	42	126	22	
5		17	13	10	166	5	7	37	22	30	110	
	1	2	3	4	5		1	2	3	4	5	
	Predicted class			Predicted class	
LR	NB	
True class	1	438	68	7	1		True class	1	337	104	26	42	5	
2	91	355	19	7	5	2	103	263	28	33	50	
3	23	87	74	33	15	3	21	58	55	56	42	
4	2	17	13	172	17	4	5	10	15	151	40	
5	1	49	6	12	138	5		20	8	11	167	
	1	2	3	4	5		1	2	3	4	5	
	Predicted class				Predicted class	
KNN	NN	
True class	1	446	52	14	2		True class	1	426	75	10	2	1	
2	134	314	18	3	8	2	78	333	37	11	18	
3	30	51	112	30	9	3	13	48	136	19	16	
4	4	6	8	191	12	4		9	20	183	9	
5	10	24	11	16	145	5		12	20	17	157	
	1	2	3	4	5		1	2	3	4	5	
	Predicted class		Predicted class	

Cubic SVM correctly classified 1,313 of the 1,650 test images, while 337 were incorrectly classified. Similarly, the least successful classifier was Fine Tree, with 55.1% accuracy. Fine Tree correctly classified 909 of the 1,650 test images and incorrectly classified 741.

Results of the proposed model

In the proposed model, for the detection of knee arthritis, feature maps obtained from both the original and Gaussian applied datasets with the DenseNet201 architecture were combined to bring together different features of the same image. After this merging process, 2,000 features were obtained for each image. Then, 600 features were selected using the NCA method to increase the performance of the proposed model. Six distinct classifiers were used to classify the optimized feature map. Table 5 shows the accuracy values of the suggested model derived from six distinct classifiers.

Table 5 Accuracy of the proposed model on different classifiers (%).

Classifiers	Accuracy (%)	
DT	53.8	
NB	64.1	
KNN	78.5	
NN	79.9	
LR	75.6	
SVM	85.0	

With an 85% success rate, the SVM classifier is the most effective classifier, according to Table 5. Neural network follows the SVM classifier at 79.9%, KNN at 78.5%, Logistic Regression at 75.6%, Naive Bayes at 64.1%, and Fine Tree at 53.8%. Since the highest performance values in the proposed model are reached in the SVM classifier, SVM is preferred as the classifier in the proposed model. The confusion matrix of the proposed model is in Table 6.

Table 6 Confusion matrix of proposed model.

SVM	
True class	1	461	51	2			
2	46	401	27	1	2	
3	4	42	162	17	7	
4		2	15	197	7	
5		8	13	4	181	
	1	2	3	4	5	
Predicted class	

Table 6 shows that the suggested model correctly identified 1,402 of the 1,650 test images, while 248 were incorrectly identified. In the Normal class, 461 of 514 test images were correctly classified, and 53 were incorrectly classified. In the Doubtful class, 401 of 477 test images were correctly classified, and 76 were incorrectly classified. In the Mild class, 162 of 232 test images were correctly classified, and 70 were incorrectly classified. In the Moderate class, 197 of 221 test images were correctly classified, and 24 were incorrectly classified. Finally, 181 of 206 test images were correctly classified in the Severe class, and 25 were incorrectly classified. Table 7 shows the suggested model’s performance measuring metrics.

Table 7 Performance metric of proposed model (%).

Classes	Accuracy	Sensitivity	Specificty	Matthews correlation coefficient	F1-score	
1-Normal	89.69	90.22	95.35	85.42	89.95	
2-Doubtful	84.07	79.56	93.37	74.10	81.75	
3-Mild	69.83	73.97	95.11	67.42	71.84	
4-Moderate	89.14	89.95	98.32	87.94	89.55	
5-Severe	87.86	91.88	98.28	88.44	89.83	

Discussion

This study will help physiotherapists evaluate patients’ condition more quickly and accurately. Advanced image processing techniques can analyze medical images such as X-rays, MRIs, or ultrasounds. In this way, injuries or disorders can be identified more accurately. The study presents results with new and advanced technology that offers higher performance than existing approaches by using data obtained from plain radiographs for automatic diagnosis of KOA. This innovative approach provides higher accuracy and reliability than existing diagnostic methods in the literature. This method increases accuracy and improves overall performance, especially in diagnosing knee OA. There are studies on the subject in the literature. These studies are presented in Table 8.

Table 8 Literature review.

Paper	Year	Method	Metrics (%)	
Zhang et al. (2020)	2020	Convolutional block attention module	Accuracy = 74.81	
Pedoia et al. (2019)	2019	U-Net CNN	AUC = 88	
Yildirim & Mutlu (2024)	2024	LBP, HOG, CNN	Accuracy = 83.6	
Tiulpin et al. (2018)	2018	CNN	AUC = 93	
Antony et al. (2016)	2016	CNN	Accuracy = 67	
Proposed model	–	Gauss, DenseNet201, NCA, SVM	Accuracy = 85	

Examining Table 8, it is evident that the suggested model outperforms the recognized models in the literature in terms of effective outcomes. However, it is possible to talk about data limitations here. Using data from different centers and increasing the number of data may affect the performance of the model. In addition, due to the lack of data sets in the literature on the subject, the proposed model was tested on a single data set. This is another limitation of our article. In the future, our aim is to conduct studies with data obtained from more and different regions.

Conclusions

KOA limits people’s comfort in living. Therefore, early detection of osteoarthritis severely impacts the patient’s comfort of living. In this article, a hybrid system was developed for early detection of the osteoarthritis stage. A high accuracy rate of 85% was obtained in the developed system. Our study accelerates research in physiotherapy and provides more in-depth analysis thanks to its ability to analyze large data sets. In this way, our model will contribute to developing new treatment methods and improving existing applications. In addition, this study demonstrated that the Gaussian denoising filter successfully descends noise in knee osteoarthritis images.

Supplemental Information

Supplemental Information 1 Raw data and code.

The authors thank the owners of the dataset for sharing their data.

Additional Information and Declarations

Competing Interests

Bilal Alatas is an Academic Editor for PeerJ.

Author Contributions

Burak Bugday conceived and designed the experiments, analyzed the data, prepared figures and/or tables, authored or reviewed drafts of the article, and approved the final draft.

Harun Bingol conceived and designed the experiments, performed the experiments, performed the computation work, prepared figures and/or tables, authored or reviewed drafts of the article, and approved the final draft.

Muhammed Yildirim conceived and designed the experiments, performed the experiments, analyzed the data, performed the computation work, authored or reviewed drafts of the article, and approved the final draft.

Bilal Alatas conceived and designed the experiments, authored or reviewed drafts of the article, and approved the final draft.

Data Availability

The following information was supplied regarding data availability:

The raw measurements are available in the Supplemental Files.

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
