# Peer review of "Enhancing knee osteoarthritis detection with AI, image denoising, and optimized classification methods and the importance of physical therapy methods"

_PeerJ Computer Science, doi:10.7717/peerj-cs.2766_

## Round 0.1 · original submission · Major Revisions

The experts have now commented on the paper's quality and you will see a couple of comments to be incorporated before we re-consider it. please carefully incorporate and consider the following points as well.

The paper does not provide sufficient information about the dataset used, such as the number of images, diversity of the dataset

While the DenseNet201 architecture was selected from eight different pre-trained models, the paper lacks a comprehensive comparison of these models' performances.

Although six classifiers were used, the paper only reports the accuracy of the SVM classifier. Providing performance metrics (e.g., precision, recall, F1-score) for all classifiers would give a more comprehensive evaluation of the system's performance and robustness.

Whats the logic behind using the Gaussian filter for denoising is mentioned.

Reviewer 1 ·

Basic reporting

I have examined the study you have carried out to detect knee osteoarthritis in detail. First, I would like to state that the focused problem is important. Original methodology is followed and promising results are presented. Many points for this problem have been successfully addressed, however there are some deficiencies.

Experimental design

A high performance is mentioned in the contribution and novelty section, but numerical values do not support this statement. When the proposed model section is examined, feature selection is performed with DenseNet201. It should be explained why these architectures were chosen although there are different architectures in the literature.

Validity of the findings

In Line 267, it is mentioned that the NCA method increases the model’s performance, but the results obtained without applying NCA in the study are not presented. How did you conclude?
When Table 1 is examined, it is seen that more than one pre-trained model was tested. However, the feature maps obtained in these models were not classified in the classifiers. Only the performances of the Densenet201 architecture on the classifiers are discussed. Adding other results will reveal the power of the proposed model more clearly.
You should express more clearly the factors or reasons that brought the model suggested in Table 6 to the fore. The statement that the experiments were carried out under the same conditions should be written. Limitations and suggestions should be included in the Discussion section. Future research directions should also be mentioned.

Additional comments

The expression “Knee osteoarthritis” has not been used in different parts of the article, and uppercase and lowercase letters have not been considered—for example, in line 22. Osteoarthritis (OA) was abbreviated in Line 18 and continued to be abbreviated in the rest of the article, for example, in Line 34. The abbreviations OA and KOA have been used interchangeably. This error should be reviewed.
The architecture names should be written in the same format throughout the article. Personal pronouns should not be used as much as possible. “Metics” in Table 6 should be corrected as “Metrics”. The spelling errors in Line 163 should be corrected. Reference used as “[h11]” in Line 267 must be corrected. All references should also be checked after this correction. Subtitle “Results obtained on pre-trained Models” should be corrected as “Results Obtained on Pre-trained Models”

Reviewer 2 ·

Basic reporting

I believe this well-written work has the potential to add to the body of knowledge on knee arthritis detection. However, the quality of the article will improve if some of the study's shortcomings are fixed.

1. The Abstract section should provide a more thorough explanation of the significance of automatic knee arthritis detection, the article's novel features, and its contributions to the literature.

2. The key results that were achieved should also be reported at the conclusion of the Abstract section.

3. The title makes reference to the connection between physical therapy and knee arthritis. In the contributions section, this statement must be emphasized.

4. The literature contains a few studies on this specific topic. It is necessary to expand these investigations. The literature review should also include a detailed discussion of the benefits and drawbacks of the pertinent studies.

5. Line 162 states that the suggested model produced competitive outcomes. It is also necessary to add the outcomes that were acquired here.

6. It is noted that the photos' noise was removed using the Gauss method. Nevertheless, the method section does not describe the Gauss method.

7. Please be very mindful of abbreviations. Explain acronyms when they are first used, using parenthesis to show the abbreviation. The abbreviation should then be used. Examine the use of OA for osteoarthritis, for instance. What is "NCA" in its extended form? In the manuscript, it is utilized straight without a lengthy version.

8. "Matlab" has to be changed to "MATLAB."

9. The full names of the writers should be added to Reference 1.

10. A single style should be used to write all the values in the tables. Values, for instance, can be expressed as two numbers following a comma.

11 Table 6 needs to be written accurately. For many publications, the "year" information in the second columns appears to be missing.

Experimental design

N/A

Validity of the findings

It is evident that the application results employ six distinct classifiers, and the outcomes of these classifiers are displayed. The method part doesn't go into detail about the classifiers.

The essay should include specifics on the Grad-cam technique. If possible, equations should be used to support the flaws listed.

The application result section provides a detailed presentation of the suggested model's outcomes. Nevertheless, the study's findings on the classifiers are only based on the Densenet201 design, ignoring other models.

A thorough discussion of the study's shortcomings is required.

---

## Round 0.2 · accepted · Accept

Thank you for your revised updated version, I'm pleased to inform you that experts are now happy with the revised updated paper. Therefore we recommend it for publication.

Reviewer 1 ·

Basic reporting

No comment

Experimental design

No comment

Validity of the findings

No comment

Reviewer 2 ·

Basic reporting

Thank you for submitting your revised manuscript. I have now had a chance to read it through, along with your responses to the points that I raised in my original review.

I am very happy with the way that you have addressed all of my comments and suggestions. The manuscript is now much improved and I am happy to recommend it for publication in its current form.

Experimental design

Thank you for submitting your revised manuscript. I have now had a chance to read it through, along with your responses to the points that I raised in my original review.

I am very happy with the way that you have addressed all of my comments and suggestions. The manuscript is now much improved and I am happy to recommend it for publication in its current form.

Validity of the findings

Thank you for submitting your revised manuscript. I have now had a chance to read it through, along with your responses to the points that I raised in my original review.

I am very happy with the way that you have addressed all of my comments and suggestions. The manuscript is now much improved and I am happy to recommend it for publication in its current form.